# Application of Core–Shell Metallic Nanoparticles in Hybridized Perovskite Solar Cell—Various Channels of Plasmon Photovoltaic Effect

**DOI:** 10.3390/ma12193192

**Published:** 2019-09-29

**Authors:** Katarzyna Kluczyk-Korch, Christin David, Witold Jacak, Janusz Jacak

**Affiliations:** 1Department of Electronics Engineering, University of Rome Tor Vergata, Via del Politecnico 1, 00133 Rome, Italy; katarzyna.kluczyk@pwr.wroc.pl; 2Institute of Condensed Matter Theory and Optics, Abbe Center of Photonics, Friedrich Schiller University of Jena, Max-Wien-Platz 1, 07743 Jena, Germany; christin.david@uni-jena.de; 3Department of Quantum Technology, Wrocław University of Science and Technology, Wyb. Wyspiańskiego 27, 50-370 Wrcław, Poland; witold.aleksander.jacak@pwr.edu.pl

**Keywords:** perovskite solar cells, plasmons, metallized cells, core–shell nanoparticles, plasmon photo-effect

## Abstract

We analyze the microscopic mechanism of the improvement of solar cell efficiency by plasmons in metallic components embedded in active optical medium of a cell. We focus on the explanation of the observed new channel of plasmon photovoltaic effect related to the influence of plasmons onto the internal cell electricity beyond the previously known plasmon mediated absorption of photons. The model situation we analyze is the hybrid chemical perovskite solar cell CH${}_{3}$NH${}_{3}$PbI${}_{3-\alpha }$Cl${}_{\alpha }$ with inclusion of core–shell Au/Si0${}_{2}$ nanoparticles filling pores in the Al${}_{2}$O${}_{3}$ or TiO${}_{2}$ porous bases at the bottom of perovskite layer, application of which improved the cell efficiency from 10.7 to 11.4% and from 8.4 to 9.5%, respectively, as demonstrated experimentally, mostly due to the reduction by plasmons of the exciton binding energy.

## 1. Introduction

An increase of the efficiency of semiconductor solar cells due to the mediation of sun-light energy absorption by surface plasmons in metallic nanoparticles has been investigated for the last few years [1,2,3,4,5,6,7,8,9]. It has been shown that metallic nanoparticles (Au, Ag, Cu) deposited on the photo-active surface of the semiconductor act as channels for solar energy transfer, resulting in a large increase of the photo-effect efficiency despite typically very low surface concentration of metallic components. The experimentally observed such plasmonic effect is usually linked with a local focusing of electric field of incident light in regions close to the curvature of metallic components and to scattering of incident light on metallic nanoparticles, which results in an increase of photon paths in a substrate semiconductor layer. However, these classical factors are not able to explain experimentally observed plasmon strengthening of photo-effect and the discrepancy between classical estimations (e.g., using Comsol system) and experimental data of efficiency increase reach one order of the magnitude [10,11]. A reason for this has been identified in quantum corrections which allow for assessment of plasmon damping when plasmons are coupled with semiconductor substrate. This coupling modifies strongly the dielectric functions of system components, both of metallic nanoparticles and locally of semiconductor, and allows for fitting the experiment. The detailed theory for this phenomenon has been developed for metallic nanoparticles deposited on solar cell surfaces consistent with experimental observations [10,12]. This theory concerns the bare metallic nanoparticles deposited on the top of a cell, i.e., when the plasmon resonance in metallic nanoparticles is negligibly shifted by the semiconductor substrate. The situation changes, however, if the metallic components are completely embedded in the semiconductor medium. Such an opportunity occurs in the case of recently advanced perovskite solar cells [13], which are manufactured by low temperature chemical engineering allowing for simple admixture of metallic components inside the photoactive layer in due of chemical procedures and layer deposition. Various preparation techniques and architectures of perovskite cells are proposed including even screen printing or ink-jet printing, when the admixing of nano-metallic components would be especially easy and controllable.

To avoid the worsening of a carriers mobility, an increase of local recombination of excitons and other perturbations of band structure, the core–shell nanoparticles consisting of metallic (Au, Ag or Cu) core coated with the insulating dielectric shell-layer (e.g., of SiO${}_{2}$) are applied in perovskite cells. The dielectric spacer separates surface plasmons in metallic core from charge carriers in the surrounding semiconductor and, on the other hand, allows for tuning of plasmon resonance via variation of insulating layer thickness and its permittivity. In perovskite cells, application of core–shell metallic components improved significantly the efficiency [13,14,15], but it has been evidenced that an unknown channel of plasmon photovoltaic effect plays here the role beyond the sun-light energy absorption step.

In the present paper, we provide the microscopic theory of plasmon photovoltaic when dielectric-coated metallic nanoparticles are completely embedded in the semiconductor medium including the influence of plasmons onto absorption of photons and on internal electricity in the cell, the latter being especially important in perovskite cells. We apply the Fermi golden rule approach to describe the plasmon photovoltaic effect. The simulation of plasmon-induced increase of the efficiency of hybrid chemical perovskite CH${}_{3}$NH${}_{3}$PbI${}_{3-\alpha }$Cl${}_{\alpha }$ cell with Au/SiO${}_{2}$ complexes has been performed in agreement with the experimental observations [13] when the plasmon induced reduction of the exciton binding energy has been taken into account. Perovskite cells are the most rapidly developing photovoltaic technology. Starting from ca. 3% efficiency in the first construction (2009), the efficiency reaches now almost 26 % (2019) (or even 30% in tandem configuration). The problem is, however, with a quick degradation of perovskite material, but, in the case of the improvement of durability of these cells, they can be the most promising for mass commercial photovoltaics because of a very cheap and easy technology. An additional increase of the efficiency by application of metallic components is thus of particular interest in perovskite cells.

## 2. Assessment of the Efficiency of Coupling between the Plasmon Dipole in Au Core of Nanoparticle and the Perovskite Medium

In the ordinary photo-effect [16], the interband transitions are confined to only vertical ones between states with almost the same momentum due to the momentum conservation and the very small momentum of sun-light photons (owing to large light velocity, *c*). Interaction with localized plasmons removes, however, the momentum conservation constraints (due to breaking of the translational symmetry) and non-vertical interband transitions of electrons can contribute to the photo-effect. The oblique inter-band transitions induced by plasmons can be accounted for via the Fermi golden rule (FGR). According to the FGR scheme [17], the probability of the inter-band transitions is proportional to the square of the matrix element of the perturbation potential between initial and final states and summed up over all initial states in the valence band and over all final states in the conduction band with only the energy conservation imposed. The essential for FGR is the form of the interaction which induces inter-band transitions and this interaction for plasmon photo-effect is different in comparison to the interaction responsible for the ordinary photo-effect.

Let us consider Au spherical nanoparticles with radius *b* coated with the dielectric layer of SiO${}_{2}$ up to the radius *a*, embedded completely in a perovskite layer—cf. Figure 1.

In the metallic core, a dipole surface plasmon is excited by incident sun-light penetrating solar cell layers. Strong surface plasmon resonance on Au nanoparticles (with radii b∈(10,100) nm, as visualized in Figure 2—for comparison, the resonances of Ag and Cu have also been shown in the figure) causes an intensive absorption of light in nanoparticles even at relatively low concentration. The sun-light energy accumulated in surface plasmon oscillations in metallic nanoparticles is quickly transferred to the surrounding semiconductor via very efficient channel caused by the coupling of semiconductor band electrons to plasmons in their near-field zone. The efficiency of this channel is governed by three competing factors: (1) the local strengthening of the electric field close to the nanoparticle curvature, (2) increase of photon penetration path in semiconductor layer due to scattering on metallic centers, and (3) admission of non-vertical inter-band electron transitions in the surrounding semiconductor. The first factor favors smaller nanoparticles with larger curvature, but the lowering of the nanoparticle radius *b* reduces the number of electrons inside the sphere, which inconveniently lowers the amplitude of the dipole plasmon (the amplitude of the dipole plasmon is proportional to $b^{3}$ [18]). The second and third factors prefer smaller nanoparticles, when the violation of the translational invariance is more important. All these tendencies lead to the complicated trade-off resulting in optimal size of metallic nanoparticles for photovoltaic applications.

Before presentation of the relevant formalism, let us comment briefly on the influence of a dielectric coating on plasmon oscillations in the metallic core–shell nanoparticle. If a metallic nanoparticle is surrounded by an infinite dielectric medium with the permittivity ε, the resonance plasmon frequency shifts, as illustrated in Figure 3. It is noticeable that the influence of the dielectric surroundings onto the plasmon resonance in metallic nanoparticles is strong—cf. Figure 3 (we determined it by Comsol simulation and by a Mie-type approach). The situation changes again if the metallic spherical nanoparticle is coated with a dielectric layer of finite thickness. The plasmon resonance also shifts and the shift is dependent on the dielectric layer thickness and of its relative permittivity. For the relative permittivity ε>1, we always deal with the red shift. The illustration of the plasmon resonance in Au and Ag spherical core shifted by the dielectric coating is presented in Figure 4 (Comsol and Mie simulation) for exemplary radius of the metallic core b=20 nm at varying dielectric layer thickness and relative permittivity of dielectric material. For thin dielectric coating (as in [13]), the plasmon modification is, however, small.

The coupling of the dipole mode of surface plasmons in metallic core of a core–shell type nanoparticle with band electrons in the nearby semiconductor we assume as the electric field of the oscillating dipole in the near-field zone (the magnetic field component vanishes in this zone), i.e., in the following form [19,20],
(1)$$E\left(R,t\right)=Re\frac{1}{\varepsilon }\left(-\frac{D(t)}{R^{3}}+\frac{3\hat{n}\left(\hat{n}\cdot D\left(t\right)\right)}{R^{3}}\right)e^{ikR},$$
where ω is the frequency of surface plasmon dipole oscillations, $D\left(t\right)=D_{0}e^{-i\omega t}$, ε is the dielectric permittivity of a medium where the metallic sphere is embedded. R is the vector from the center of the sphere with radius *b* to an arbitrary point outside the metallic core, i.e., R≥b, $\hat{n}=\frac{R}{R}$ (cf. Figure 1). The dipole of surface plasmon in the spherical metallic core is pinned to the center of the nanosphere (the origin of the reference frame for this system). The exponent, $e^{ikR}$, is caused by the retardation effect when the retarded Fourier component, $e^{-i\omega (i-\frac{R}{v})}$, is rewritten as the ordinary Fourier component, $e^{-i\omega t}$, multiplied by the factor, $e^{i\omega \frac{R}{v}}=e^{ikR}$, where $k=\frac{\omega }{v}$ and *v* is the averaged velocity of light on the distance *R*. The dynamical electric field given by Equation (Equation 1) causes the perturbation potential for the band electrons in the semiconductor, for R>a. This perturbation potential (acc. to Equation (Equation 1)) attains the following form (neglecting retardation on small distances):(2)$$\begin{array}{c} w\left(R,t\right)=e\psi \left(R,t\right)=\frac{e}{\varepsilon R^{2}}\hat{n}\cdot D_{0}sin\left(\omega t+\alpha \right)=w^{+}e^{i\omega t}+w^{-}e^{-i\omega t} \end{array}$$
because $-\nabla_{R}w\left(R,t\right)=eE\left(R,t\right)$ and $\nabla_{R}R=\hat{n}$. We choose the term with +, i.e., $w^{+}={\left(w^{-}\right)}^{*}=\frac{e}{\varepsilon R^{2}}\frac{e^{i\alpha }}{2i}\hat{n}\cdot D_{0}$ because we want to describe the emission of the energy form the oscillating dipole (the term with–describes the symmetrical absorption by the plasmon dipole of the energy from the semiconductor).

Upon the scheme of FGR [17], the probability per time unit of interband transitions of electrons in the semiconductor induced by plasmon oscillations in the metallic nanoparticle attains the form:(3)$$w\left(k_{1},k_{2}\right)=\frac{2\pi }{\hbar }{\left|<k_{1}\left|w^{+}\right|k_{2}>\right|}^{2}\delta \left(E_{p}\left(k_{1}\right)-E_{n}\left(k_{2}\right)+\hbar \omega \right),$$
where $k_{1}$ and $k_{2}$ denote initial and final quantum states in the semiconductor band system. For simplicity, we assume the band Bloch states in the valence and conduction bands of the semiconductor as planar waves within the effective mass approximation, i.e., $\Psi_{k}=\frac{1}{{\left(2\pi \right)}^{3/2}}e^{ik\cdot R-iE_{n(p)}\left(k\right)t/\hbar }$, $E_{p}\left(k\right)=-\frac{\hbar^{2}k^{2}}{2m_{p}^{*}}-E_{g},E_{n}\left(k\right)=\frac{\hbar^{2}k^{2}}{2m_{n}^{*}}$. By the indices, *n* and *p*, we denote electrons from the conduction and valence bands, respectively. $E_{g}$ denotes here the forbidden gap between the valence and conduction bands of the semiconductor.

First, we must calculate the matrix element in Equation (Equation 3). This matrix element has the form,
(4)$$<k_{1}\left|w^{+}\right|k_{2}>=\frac{1}{{\left(2\pi \right)}^{3}}\int d^{3}R\frac{e}{\varepsilon 2i}e^{i\alpha }\hat{n}\cdot D_{0}\frac{1}{R^{2}}e^{-i(k_{1}-k_{2})\cdot R},$$
where $k_{1(2)}$ is the band electron pseudo-momentum (in the first Brillouin zone) for the valence (conduction) band. In explicit form, Equation (Equation 4) gains the following shape, with an integral with respect to dR taken from *a*, at which the semiconductors meet with the dielectric shell of the metallic core,
(5)$$<k_{1}\left|w^{+}\right|k_{2}>=\frac{-1}{{\left(2\pi \right)}^{3}}\frac{ee^{i\alpha }}{\varepsilon }D_{0}cos\Theta \left(2\pi \right)\int_{a}^{\infty }dR\frac{1}{q}\frac{d}{dR}\frac{sinqR}{qR}=\frac{1}{{\left(2\pi \right)}^{2}}\frac{ee^{i\alpha }}{\varepsilon }\frac{D_{0}\cdot q}{q^{2}}\frac{sinqa}{qa},$$
where $q=k_{1}-k_{2}$. The next step is the summation over all initial and final states in the valence and conduction band, respectively. The overall probability of the interband transition thus attains the form,
(6)$$\delta w=\int d^{3}k_{1}\int d^{3}k_{2}\left[f_{1}\left(1-f_{2}\right)w\left(k_{1},k_{2}\right)-f_{2}\left(1-f_{1}\right)w\left(k_{2},k_{1}\right)\right],$$
where $f_{1},f_{2}$ are the temperature dependent distribution functions (Fermi–Dirac distribution functions for electrons) for the initial and final states, respectively. At room temperatures, when $f_{2}≃0$ and $f_{1}≃1$—i.e., the conduction band is almost empty and the valence band is almost fully occupied, we obtain
(7)$$\delta w=\int d^{3}k_{1}\int d^{3}k_{2}w\left(k_{1},k_{2}\right).$$

The integrals in the above formula can be performed analytically, which allows for the compact expression,
(8)$$\begin{array}{c} \delta w=\frac{4}{3}\frac{\mu^{2}\left(m_{n}^{*}+m_{p}^{*}\right)2\left(\hbar \omega -E_{g}\right)e^{2}D_{0}^{2}}{\sqrt{m_{n}^{*}m_{p}^{*}}2\pi \hbar^{5}\varepsilon^{2}}\int_{0}^{1}dx\frac{sin^{2}\left(xa\xi \right)}{{\left(xa\xi \right)}^{2}}\sqrt{1-x^{2}}\\ =\frac{4}{3}\frac{\mu^{2}}{\sqrt{m_{n}^{*}m_{p}^{*}}}\frac{e^{2}D_{0}^{2}}{2\pi \hbar^{3}\varepsilon^{2}}\xi^{2}\int_{0}^{1}dx\frac{sin^{2}\left(xa\xi \right)}{{\left(xa\xi \right)}^{2}}\sqrt{1-x^{2}}, \end{array}$$
where $\mu =\frac{m_{n}^{*}m_{p}^{*}}{m_{n}^{*}+m_{p}^{*}}$ is the reduced mass. We have introduced here the parameter $\xi =\frac{\sqrt{2\left(\hbar \omega -E_{g}\right)\left(m_{n}^{*}+m_{p}^{*}\right)}}{\hbar }$, which allows the definition of two limiting cases for the coated nanoparticle radius *a*, when ξa≪1 (small nanoparticles) and ξa≫1 (large nanoparticles). Let us remind that *a* is the external radius of the dielectric coating of smaller metallic nanoparticle core with surface plasmon. We finally obtain (again in the analytical form),
(9)$$\begin{array}{c} \delta w=\left\{\begin{array}{c} \frac{4}{3}\frac{\mu \sqrt{m_{n}^{*}m_{p}^{*}}\left(\hbar \omega -E_{g}\right)e^{2}D_{0}^{2}}{\hbar^{5}\varepsilon^{2}},\;\mathrm{for}\;a\xi \ll 1,\\ \frac{4}{3}\frac{\mu^{3/2}\sqrt{2}\sqrt{\hbar \omega -E_{g}}e^{2}D_{0}^{2}}{a\hbar^{4}\varepsilon^{2}},\;\mathrm{for}\;a\xi \gg 1. \end{array}\right. \end{array}$$

The above formula, i.e., Equation (Equation 8) and the explicit forms in limiting situations given by Equation (Equation 9), formulate the generalization of the ordinary photo-effect. The latter describes the probability of electron inter-band transitions induced by a plane wave of incident photons, expressed as follows [16]:(10)$$\begin{array}{c} \delta w_{0}=\frac{4\sqrt{2}}{3}\frac{\mu^{5/2}e^{2}}{m_{p}^{*2}\omega \varepsilon \hbar^{3}}\left(\frac{\varepsilon E_{0}^{2}V}{8\pi \hbar \omega }\right){\left(\hbar \omega -E_{g}\right)}^{3/2}. \end{array}$$

The number of photons with the energy ℏω corresponding to the e-m wave with the electric field component amplitude $E_{0}$, contained in the volume *V* equals $\left(\frac{\varepsilon E_{0}^{2}V}{8\pi \hbar \omega }\right)$. Using this expression, one can rewrite the formula for the ordinary photo-effect, as the probability of the single photon absorption by the semiconductor per time unit, which gains the following, more familiar form [16],
(11)$$q_{0}=\delta w_{0}{\left(\frac{\varepsilon E_{0}^{2}V}{8\pi \hbar \omega }\right)}^{-1}=\frac{4\left(4\right)\sqrt{2}}{3}\frac{\mu^{5/2}e^{2}}{m_{p}^{*2}\omega \varepsilon \hbar^{3}}{\left(\hbar \omega -E_{g}\right)}^{3/2},$$
(factor (4) corresponds here to the spin degeneracy of band electrons).

In the analogous manner, one can normalize the probability of energy absorption in the semiconductor via mediation of surface plasmons per single incident photon. This probability normalized per single photon we denote as $q_{m}$, which equals the product of δw (given by Equation (Equation 9)) and the number, *N*, of metallic nanoparticles divided by the photon density,
(12)$$q_{m}=\beta N\delta w{\left(\frac{\varepsilon E_{0}^{2}V}{8\pi \hbar \omega }\right)}^{-1}.$$
We have introduced here an additional phenomenological factor, β. This factor collectively accounts for all effects not directly included in the above derivation. Such not-included effects are e.g., surface properties like surface recombination of excitons, which can reduce the coupling strength and the transition probability.

## 3. Damping of Plasmons and the Plasmon Energy Flow to a Nearby Semiconductor

The energy out-flow from plasmons to the nearby semiconductor causes their strong damping. The damping of plasmons is thus a measure of the efficiency of plasmon-mediated absorption of photons (Table 1 and Table 2).

To assess it quantitatively, let us assume that the energy in-flow to the nearby semiconductor band system, A, is equal to the out-flow of energy of plasmons oscillating in metallic nanoparticles. Considering the gradual damping of plasmons according to the exponential dipole amplitude lowering, $D_{0}\left(t\right)=D_{0}e^{-t/\tau^{\prime }}$, we can estimate the transferred energy by the following formula:(13)$$A=\beta \int_{0}^{\infty }\delta w\hbar \omega dt=\beta \hbar \omega \delta w\tau^{\prime }/2=\{\begin{array}{c} \frac{2}{3}\frac{\beta \omega \tau^{\prime }\mu \sqrt{m_{n}^{*}m_{p}^{*}}\left(\hbar \omega -E_{g}\right)e^{2}D_{0}^{2}}{\hbar^{4}\varepsilon^{2}},\;\mathrm{for}\;a\xi \ll 1,\\ \frac{2}{3}\frac{\beta \omega \tau^{\prime }\mu^{3/2}\sqrt{2}\sqrt{\hbar \omega -E_{g}}e^{2}D_{0}^{2}}{a\hbar^{3}\varepsilon^{2}},\;\mathrm{for}\;a\xi \gg 1. \end{array}$$
ε is the permittivity of the perovskite, which is ca. 5 (by $\varepsilon_{1}$, we denote the permittivity of the dielectric coating of the metallic nanosphere; for SiO${}_{2}$, it is ca. 3.9).

In the above formula, the symbol $\tau^{\prime }$ indicates the damping time-rate of surface plasmons coupled to the electrons in the semiconductor substrate. Special attention should be paid to two limiting situations, ξa≪1 (small metallic nanoparticles, with radius not larger than 2–3 nm), and ξa≫1 (large nanoparticles, of a radius larger than ca. 5 nm). To be specific, let us assess the parameter ξ for considered by us perovskite solar cell. Taking into account the material parameters for perovskite as listed in the Table 3, we find
$$\xi =\frac{\sqrt{2\left(\hbar \omega -2.56\times 10^{-19}\right)\left(18.2\times 10^{-31}\right)}}{\hbar }\left[1/m\right],$$
where $\hbar =1.05\times 10^{-34}$ J·s; ω is plasmon circular frequency in units 1/s.

Via the direct comparison of A given by Equation (Equation 13) with the initial energy of the plasmon, which has next been transferred step-by-step to the semiconductor, one can find the damping rate, $\frac{1}{\tau^{\prime }}$, for plasmons in the considered case. The initial energy of plasmons equals [18], $B=\frac{D_{0}^{2}}{2\varepsilon b^{3}}$ (it is important here that the radius *b* is from the metallic core). Assuming that A=B, we arrive with the formula for the damping rate for plasmons:(14)$$\frac{1}{\tau^{\prime }}=\left\{\begin{array}{c} \frac{4\beta \omega \mu \sqrt{m_{n}^{*}m_{p}^{*}}\left(\hbar \omega -E_{g}\right)e^{2}b^{3}}{3\hbar^{4}\varepsilon },\;\mathrm{for}\;a\xi \ll 1,\\ \frac{4\beta \omega \mu^{3/2}\sqrt{2}\sqrt{\hbar \omega -E_{g}}e^{2}b^{2}}{3\hbar^{3}\varepsilon },\;\mathrm{for}\;a\xi \gg 1. \end{array}\right.$$
By $\tau^{\prime }$, we denote here a large damping of plasmons due to the energy transfer to the semiconductor substrate highly exceeding the internal damping, characterized by τ, the latter due to the scattering of electrons inside the metallic nanoparticle [18] ($\frac{1}{\tau }\ll \frac{1}{\tau^{\prime }}$).

For nanospheres of Au/SiO${}_{2}$ embedded inside the perovskite layer, we obtain for $\omega =\omega_{1}$ (Mie self-frequency of plasmon),
(15)$$\frac{1}{\tau^{\prime }\omega_{1}}=\left\{\begin{array}{c} 0.54\beta {\left(\frac{b[nm]}{1[nm]}\right)}^{3}\frac{\mu }{m}\frac{\sqrt{m_{n}^{*}m_{p}^{*}}}{m},\;\mathrm{for}\;a\xi \ll 1,\\ 0.165\beta {\left(\frac{b[nm]}{1[nm]}\right)}^{2}{\left(\frac{\mu }{m}\right)}^{3/2},\;\mathrm{for}\;a\xi \gg 1, \end{array}\right.$$
for carriers in perovskite, $m_{n}^{*}≃1\;m$, $m_{p}^{*}≃1\;m$, *m* is the bare electron mass, $\mu =\frac{m_{n}^{*}m_{p}^{*}}{m_{n}^{*}+m_{p}^{*}}=m/2$ and $E_{g}=1.6$ eV, $\hbar \omega_{1}=2.72$ eV, ε∼5. For these parameters and nanospheres with the radius *a* in the range of 10–100 nm, the lower case of Equation (Equation 15) applies (at $\omega =\omega_{1}$).

In another scenario (as in the case of a solar cell metallically modified) when the out-flow of the plasmon energy is recovered by the continuous income from the sun-light, one can consider the energy-balanced regime. When all of the incoming to plasmon energy of the monochromatic ω e-m wave of incident photons is transferred to the semiconductor via the plasmon-electrons coupling channel, we deal with the stationary state of a driven and damped oscillator for plasmons. Hence, the frequency of plasma oscillations equals the driven electric field frequency, ω, of the incident e-m wave of photons. Because of an instant leakage of the plasmon energy in the near field-zone to the semiconductor substrate, this large damping of plasmon causes a red-shift and a widening of the resonance, as for a conventional damped and driven oscillator.

The damping of plasmons causes a red-shift of the resonance and reduces the resonance amplitude, which in turn allows for the accommodation of the balance of energy transfer to the semiconductor with the incident sun-light e-m wave energy intensity (defined by its electric field amplitude $E_{0}$) at the frequency ω. Within this damped and driven oscillator model, the accommodated amplitude of plasmon oscillations $D_{0}\left(\omega \right)$ is constant in time and shaped by the formula, $f\left(\omega \right)=\frac{1}{\sqrt{{\left(\omega_{1}^{2}-\omega^{2}\right)}^{2}+4\omega^{2}/\tau^{{}^{\prime }2}}}$. The extreme of red-shifted resonance is attained at $\omega_{m}=\omega_{1}\sqrt{1-2{\left(\omega_{1}\tau^{\prime }\right)}^{-2}}$ with the corresponding amplitude $∼\tau^{\prime }/\left(2\sqrt{\omega_{1}^{2}-\tau^{{}^{\prime }-2}}\right)$. In the case of the described energy transfer balance, one obtains according to Equation (Equation 9),
(16)$$q_{m}=\left\{\begin{array}{c} \beta C_{0}\frac{128}{9}\pi^{2}a^{3}\frac{\mu \sqrt{\mu_{n}^{*}\mu_{p}^{*}}}{m^{2}}\left(\hbar \omega -E_{g}\right)\frac{e^{6}n_{e}^{2}\omega }{\hbar^{4}\varepsilon^{3}}f^{2}\left(\omega \right),\;\mathrm{for}\;a\xi \ll 1,\\ \beta C_{0}\frac{128}{9}\sqrt{2}\pi^{2}a^{2}\frac{\mu^{3/2}}{m^{2}}\sqrt{\hbar \omega -E_{g}}\frac{e^{6}n_{e}^{2}\omega }{\hbar^{3}\varepsilon^{3}}f^{2}\left(\omega \right),\;\mathrm{for}\;a\xi \gg 1, \end{array}\right.$$
where $f\left(\omega \right)=\frac{1}{\sqrt{{\left(\omega_{1}^{2}-\omega^{2}\right)}^{2}+4\omega^{2}/\tau^{{}^{\prime }2}}}$ corresponds to the amplitude factor for the driven damped oscillator and $D_{0}=\frac{e^{2}n_{e}E_{0}4\pi b^{3}}{3m}f\left(\omega \right)$ (in Equation (Equation 9)); the amplitude of the electric field, $E_{0}$, in the incident e-m wave is next ruled out from Equation (Equation 16) due to normalization per single photon as in Equation (Equation 12); $C_{0}=\frac{N4/3\pi b^{3}}{V}$, *V* is the volume of the semiconductor (V=SH, *S* is the cell surface, *H* is the thickness of perovskite layer), *N* is the number of metallic nanospheres.

The ratio, $\frac{q_{m}}{q_{0}}$, revealing the advantage of the plasmon mediated photo-effect over the ordinary photo-effect can be expressed as follows:(17)$$\frac{q_{m}}{q_{0}}=\left\{\begin{array}{c} \frac{4\sqrt{2}\pi^{2}a^{3}\beta C_{0}\sqrt{m_{n}^{*}m_{p}^{*}}{\left(m_{p}^{*}\right)}^{2}e^{4}n_{e}^{2}\omega^{2}f^{2}\left(\omega \right)}{3\mu^{3/2}m^{2}\sqrt{\hbar \omega -E_{g}}\hbar \varepsilon^{2}},\;\mathrm{for}\;a\xi \ll 1,\\ \frac{8\pi^{2}a^{2}\beta C_{0}{\left(m_{p}^{*}\right)}^{2}e^{4}n_{e}^{2}\omega^{2}f^{2}\left(\omega \right)}{3\mu m^{2}\left(\hbar \omega -E_{g}\right)\varepsilon^{2}},\;\mathrm{for}\;a\xi \gg 1. \end{array}\right.$$

One can estimate the photo-current in the case of a semiconductor photodiode with metallic plasmonic components including here only enhancement of the absorption of photons. This photo-current is given by $I^{\prime }=\left|e\right|N_{p}\left(q_{0}+q_{m}\right)A$, where $N_{p}$ is the number of incident photons and $q_{0}$ and $q_{m}$ are the probabilities of single photon absorption in the ordinary photo-effect [16] and of single photon absorption mediated by the presence of metallic nano-spheres, respectively, as derived in the previous paragraph. $A=\frac{\tau_{f}^{n}}{t_{n}}+\frac{\tau_{f}^{p}}{t_{p}}$ is the amplification factor ($\tau_{f}^{n(p)}$ is the annihilation time of both sign carriers, $t_{n(p)}$ is the drive time for carriers (the time of traversing the distance between the electrodes)). Note that we included simultaneously the absorption of photons directly in the semiconductor, $q_{0}$, and that one due to mediation of plasmons, $q_{m}$. From the above formulae, it follows that (here $I=I^{\prime }\left(q_{m}=0\right)$, i.e., the photo-current without metallic modifications),
(18)$$\frac{I^{\prime }}{I}=1+\frac{q_{m}}{q_{0}},$$
where the ratio $q_{m}/q_{0}$ is given by Equation (Equation 17).

Note that collective reflection–interference type corrections worsening light absorption are instead not strong for the considered low densities of metallic coverings and nano-sphere sizes well lower than the resonant wave-length, though for larger concentrations and larger nano-sphere sizes, would play a stronger reducing role (reflecting incident sun-light photons) [3,5].

## 4. Reducing of the Binding Energy of Excitons in the Perovskite Cell Due to Coupling to Plasmons in Metallic Core–Shell Nanoparticles

Besides the plasmon influence on the photon absorption in the nearby semiconductor, plasmonic elements can also induce changes in the internal electricity of the cell. According to the experimental demonstration of the latter effect, a strong reduction of the binding energy of excitons in perovskite cells, from 100 meV to 30 meV, has been reported due to core–shell Au/SiO${}_{2}$ nanoparticles embedded in the perovskite basis of porous Al${}_{2}$O${}_{3}$ or porous TiO${}_{2}$ in the chemical hybrid perovskite cell [13]. The reduction of the exciton binding energy has been measured by photo-luminescence methods by comparison of the signal form the metallically doped and reference structures without nanoparticles on several samples. Authors of Ref. [13] argue that the observed increase in overall efficiency of metallically modified solar cells (from 10.7% to ca. 11.4% for Al${}_{2}$O${}_{3}$ and from 8.5% to 9.7% for TiO${}_{2}$) may instead be attributed to the reduction of the exciton binding energy close to porous basis rather than to strengthening of photon absorption. Lower binding energy facilitating separation of opposite sign carries and enhances photo-current. This might be realistic if one takes into account that, in perovskite, the effective mass of electron and holes are practically equal to free electron mass, which strongly reduces plasmon mediated light absorption in comparison to e.g., Si cells with light electrons and light holes. Moreover, the larger forbidden gap in perovskite cuts off a significant part of the sun-light spectrum. Therefore, the reducing of the exciton binding energy would be more important in the case of perovskite chemical cell, especially when metallic core–shell nanoparticles Au/SiO${}_{2}$ (with radius b≃40 nm and radius a≃48 nm, as applied in Ref. [13]) are located in pores of the Al${}_{2}$O${}_{3}$ or TiO${}_{2}$ (with concentration of ca. 1 wt %).

To reproduce the effect described above, let us emphasize once more that coupling of surface dipole plasmons in metallic nanoparticles perturbing the electron band system in the nearby semiconductor causes interband transitions of electrons, which can be non-vertical due to the specific form of the perturbation potential, Equation (Equation 2). We have already noticed that the matrix element of this potential between Bloch functions of band electrons, Equation (Equation 4), is not diagonal in pseudo-momentum, i.e., non-vertical interband transitions are admitted. This is in contrast to the ordinary photo-effect, when the perturbation of electrons in a semiconductor was induced by the vector potential of e-m wave entering the kinematical momentum of band electrons [16]. For planar e-m waves of incident photons, this vector potential has the form $∼e^{ip\cdot r}$, where $\hbar p=\frac{\hbar \omega }{c}$ is the momentum of incident photon (ℏω is the energy of this photon, and *c* is light velocity). The matrix element between Bloch functions is in this case diagonal in electron pseudo-momenta because the integral of $e^{i(k_{1}-k_{2}+p)\cdot r}$ gives the Dirac delta, $\delta (k_{1}-k_{2}+p)$, which expresses the momentum conservation. Due to a large value of *c*, the momentum of photons is negligibly small in comparison to pseudo-momenta of electrons in the Brillouin zone of the semiconductor, and only transitions between the same k are admitted (the vertical transitions). Plasmons perturb the band electrons by the different potential—Equation (Equation 2), which has an infinite number of spatial Fourier components, resulting in nonzero transition probability between different $k_{1}$ and $k_{2}$ electron states (non-vertical transitions).

This fact causes some important consequence also with regard to the stability of the exciton formed by excited electrons to the conduction band and the hole left in the valence band. The conventional assessment of binding energy of this exciton is assumed according to an effective atom Bohr-like model, for which the binding energy is given by the formula:(19)$$E\left(n\right)=-\frac{\mu R_{H}}{m\varepsilon^{2}n^{2}},$$
where $\mu =\frac{m_{n}^{*}m_{p}^{*}}{m_{n}^{*}+m_{p}^{*}}$ is the reduced mass of electron and hole, *m* is the mass of free electron, $R_{H}=13.6$ eV is the Rydberg constant, ε is the permittivity of the semiconductor, and *n* enumerates the ground (n=1) and excited states of the exciton. This simplified model formula is, however, assumed for the ordinary photo-effect, when the pseudo-momenta of electron and hole are the same. In a plasmon mediated photo-effect, this momenta can be different. An exciton with nonzero relative momentum, $\hbar k_{1}-\hbar k_{2}=\hbar q$, has smaller binding energy, reduced approximately by $\frac{\hbar^{2}q^{2}}{2\mu }$. This is caused by opposite momenta of electron and hole components of the exciton, which tend to disrupt the weakly bound pair. This eventually leads to the reduction of the binding energy of the exciton.

The Formula (Equation 3) gives the probability (per time unit) of transition between $k_{1}$ and $k_{2}$ states induced by plasmon. Thus, the averaged reduction of the binding energy of excitons can be assessed according to the formula:(20)$$\Delta E=\int d^{3}k_{1}\int d^{3}k_{2}\frac{\hbar^{2}q^{2}}{2\mu }w\left(k_{1},k_{2}\right),$$
with $w\left(k_{1},k_{2}\right)=\frac{2\pi }{\hbar }{\left|<k_{1}\left|w^{+}\right|k_{2}>\right|}^{2}\delta \left(E_{p}\left(k_{1}\right)-E_{n}\left(k_{2}\right)+\hbar \omega \right)$ and $<k_{1}\left|w^{+}\right|k_{2}>=\frac{1}{{\left(2\pi \right)}^{2}}\frac{ee^{i\alpha }}{\varepsilon }\frac{D_{0}\cdot q}{q^{2}}\frac{sinqa}{qa}$, where $q=k_{1}-k_{2}$ (cf. Equation (Equation 5), $D_{0}$ is defined below Equation (Equation 16)). It is easy to notice that the above integral is convergent (due to sinc function properties) and positive, which explains the observed [13] phenomenon of reduction of the exciton binding energy in the range of a near-field zone of the plasmon in metallic nanoparticles. One can roughly estimate this contribution for perovskite CH${}_{3}$NH${}_{3}$PbI${}_{3-\alpha }$Cl${}_{\alpha }$ cells with the elementary cell axes, a=b≃0.85 nm, c≃0.45 nm, and the refraction index 2.5–3. The estimation of the exciton binding energy gives thus ca. 100 meV (at $\frac{\mu }{m\varepsilon^{2}}∼8\times 10^{-3}$), as reported in [13]. For the averaged q∼0.075×2π/l (where l∼0.5 nm is taken to estimate the size of the Brillouin zone, and 0.075 taken as the averaged size of *q* on the Brillouin zone scale), the reducing of the exciton binding energy (at μ=0.5 m, as in perovskite) is ΔE∼70 meV, as measured by photoluminescence methods in the metallized perovskite cell [13].

We see that plasmons cause modification both of photon absorption and of local electricity inside the photovoltaic cell metallically modified. Both these channels influence the overall efficiency of the cell with the final value being the subject of the trade-off in complicated dependence on material parameters and metallic nanoparticle deposition type. In the case of a hybrid chemical perovskite cell, the location of metallic core–shell nanoparticles in pores of the porous basis of the cell favors the second channel in agreement with the measurement of giant reduction of the exciton binding energy and related overall cell efficiency enhancement [13]. In Figure 5, we illustrate the microscopic quantum assessment of the efficiency of plasmon-induced absorption of light in the hybrid perovskite cell metallized with Au/SiO${}_{2}$ nanoparticles (bottom panels) in comparison with the same effect in Si cell (upper panels). It is evident that this channel of the plasmon effect in perovskite cell is ineffective. However, the second channel of plasmon influence (originally suggested phenomenologically by experimenters [13] and microscopically explained above) appears to be of primary importance in considered perovskite cells. This channel does not concern photon absorption strengthening by plasmons (which is at least poor in perovskite for used metallic components) but corresponds to a strong decrease of binding energy of excitons excited by the plasmon potential in its near field-zone. The lowering of exciton binding energy results in the easier dissociation of excitons at the surface of porous basis of the perovskite layer and in the eventual increase of photo-current. The experimentally observed significant increase of the photo-current (but not of the voltage) and the cell efficiency growth induced by metallic components is presented in Figure 6. The difference with the plasmon effect with the dominating channel of the absorption growth (as in the case of Si cell covered by metallic nanoparticles on the cell top) is visible by comparison of the experimental data in Figure 6 and Figure 7. The absorption channel causes an increase in both current and voltage (as in a Si case—Figure 7b,c), whereas the reducing of exciton binding energy only strengthens the photo-current (Figure 6c).

The experiment [13] and its microscopic explanation thus shed a new light on plasmons mediated efficiency enhancement of solar cells not known previously and they show more rich ways and new levers for optimization of the plasmon photovoltaic effect.

The hybrid perovskite cells differ considerably compared to conventional p−n junction-type cells, like Si-based cells or other semiconductor cells (CIGS (copper indium gallium selenide), GaAs and other). In the latter, cells’ photovoltaic excitons are instantly dissociated in the junction region because of relatively high junction voltage (e.g., in Si of ca. 0.7 V) readily overcoming the binding energy of excitons. Separated there, electrons and holes travel in opposite directions to electrodes. Another situation is, in exciton-type cells, chemical, plastic or perovskite ones, which operate without any p−n junction, and the dissociation of localized or itinerant excitons undertaken on the charge absorber surface (for electrons [13] or holes [14]), operating due to the difference of the bend edges at the interface between photo-active material and absorber. In chemical dye cells, photo-active molecules are excited by incident photons and electrons outflow from these molecules to an adjacent large gap semiconductor, TiO${}_{2}$, due to the convenient slope of energy levels at the interface. The close contact of dye molecules with a large effective surface of porous TiO${}_{2}$ is important here. The electrolyte next restores the initial state of dye by the re-dox cycle. In perovskite cells, the photo-active material is the mid forbidden gap (ca. 1.6 eV) semiconductor CH${}_{3}$NH${}_{3}$PbI${}_{3}$ (It can be substituted by Cl, which slightly changes elementary cell dimension and bend parameters as well as the chemistry of layer preparation) in which excitons are created by incident photons or by plasmons mediating sun-light absorption. These excitons dissociate at the interface with a large gap semiconductor, TiO${}_{2}$ or Al${}_{2}$O${}_{3}$ (being the electron absorbers due to the convenient slope of conduction band edges at the interface with the perovskite). A very important role is played by the time of the diffusion of excitons to the active interface because diffusion that is too long increases recombination of excitons, reducing eventual photo-current. To struggle with such a parasitic effect, the thin perovskite layers are applied (optimal are of ca. 290 nm of depth [14]) and porous bases for the perovskite layer are constructed, increasing the surface and range of the interface. All this is the subject of optimization and trade-off of various competing factors, including also metallization by plasmon nanoparticles reducing exciton binding energy and, in this way, accelerating their dissociation and increasing eventual photo-current. To optimize plasmon influence on the dissociation process, the localization of metallic nanoparticles in a close vicinity of the interface is convenient. However, the great obstacle that blocks practical utilization of perovskite cells, invincible yet, is their chemical instability and short time of degradation if exposed to atmospheric conditions. A protection by glass or plastic layers complicates technology and is, in practice, insufficient to assure durability comparable to Si cells. Various methods are proposed to increase the durability of perovskite cells, e.g., fluorination by the plasma treatment [21]. Perovskite cells exposed to the fluorination process showed very good stability against water, light, and oxygen without any impact on the morphology and electrical properties of the pristine cell. Such advances in preparation techniques show that perovskite cells might be a strong factor in the renewable green energy market in the near future.

## 5. Conclusions

By application of the quantum Fermi golden rule scheme, we have demonstrated that the efficiency of the energy transfer channel between the surface plasmon oscillations in a metallic nanoparticles and a nearby semiconductor depends on parameters of both deposited metallic particles (their radius, deposition type and concentration and material) and on semiconductor parameters (energy gap, and effective masses of electrons and holes in the conduction and valence bands, respectively). The formula found by us, which generalizes the ordinary photo-effect onto the plasmon mediated one, agrees well with the experimental measurements in a laboratory photo-diode configuration (for Si, CIGS, GaAs). We have analyzed in more detail the hybrid perovskite CH${}_{3}$NH${}_{3}$PbI${}_{3-\alpha }$Cl${}_{\alpha }$ cell with core–shell Au/SiO${}_{2}$ located with a concentration of ∼1 wt % in pores of perovskite basis of porous TiO${}_{2}$ or Al${}_{2}$O${}_{3}$. The measured ratio of absorption increase in such setup with and without metallic nano-components is compared with the theoretically predicted scenario and with conventional metallized Si cells. We confirmed the relatively poor increase of the absorption rate due to Au/SiO${}_{2}$ components, much lower than for Si cells with similar size and density of metallic elements deposited on the top of the cell. The difference is a consequence of inconveniently large masses of carriers in perovskite in comparison to Si and of a larger forbidden gap in perovskite cutting out the red and infrared part of the solar spectrum and causing a mismatch with the Au nanoparticle-mediated absorption spectrum, additionally red shifted by the dielectric coating (the latter is not large for thin coating). For perovskite cells, Ag, Cu, Fe, or even Mg, Al metallic nanoparticles, for example, with higher-energy plasmon resonances would be more effective, which would be appropriately red shifted by thick enough dielectric coating or by multiple shell mixed metallic structure.

However, the metallized by Au/SiO${}_{2}$ particles perovskite cell exhibits a ca. 16% increase of the overall efficiency, mostly due to a large increase of the photo-current. We confirmed the explanation of the authors of the original experiment that the reason for this increase is not related with the absorption enhancement but with the reduction of the binding energy of photo-excitons. This phenomenon we explained by the inclusion of the non-vertical inter-band hopping of electrons induced by the coupling to plasmons in core–shell particles in the near-field regime. We assessed the binding energy reduction of excitons related to nonzero relative momentum of exciton electron-hole components excited by plasmon, which did not occur in the ordinary photo-effect, when the relative momentum of photo-exciton components was always zero. We found the averaged value of relative momentum by plasmon excited excitons by application of the Fermi golden rule. This plasmon effect is a new way to increase the efficiency of solar cells via metallic components which modify not only the absorption of photons but also the local electricity inside the perovskite cell.

## Figures and Tables

**Figure 1 materials-12-03192-f001:**
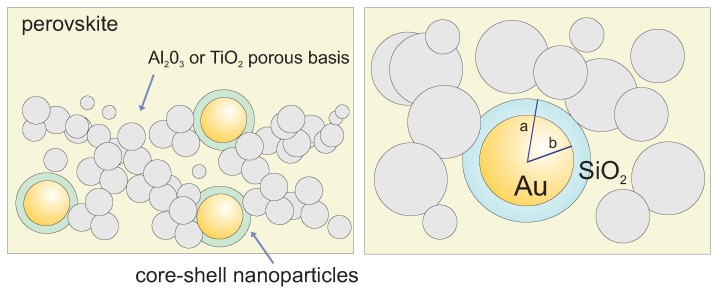
The spherical Au nanoparticles with radius b∼40 nm coated with the dielectric layer of SiO${}_{2}$ of final radius a∼48 nm are embedded in the perovskite layer of the chemical perovskite solar cell. Core–shell plasmonic nanoparticles are located in pores of Al${}_{2}$O${}_{3}$ or Ti0${}_{2}$ porous basis with a low density of particles, $n=\frac{N}{V}=\frac{N}{SH}$, corresponding to ca. 1 wt % of porous basis, where *N* is the number of plasmonic nanoparticles, *S* is the surface of the perovskite layer and *H* is its depth (experiment in Ref. [13]).

**Figure 2 materials-12-03192-f002:**
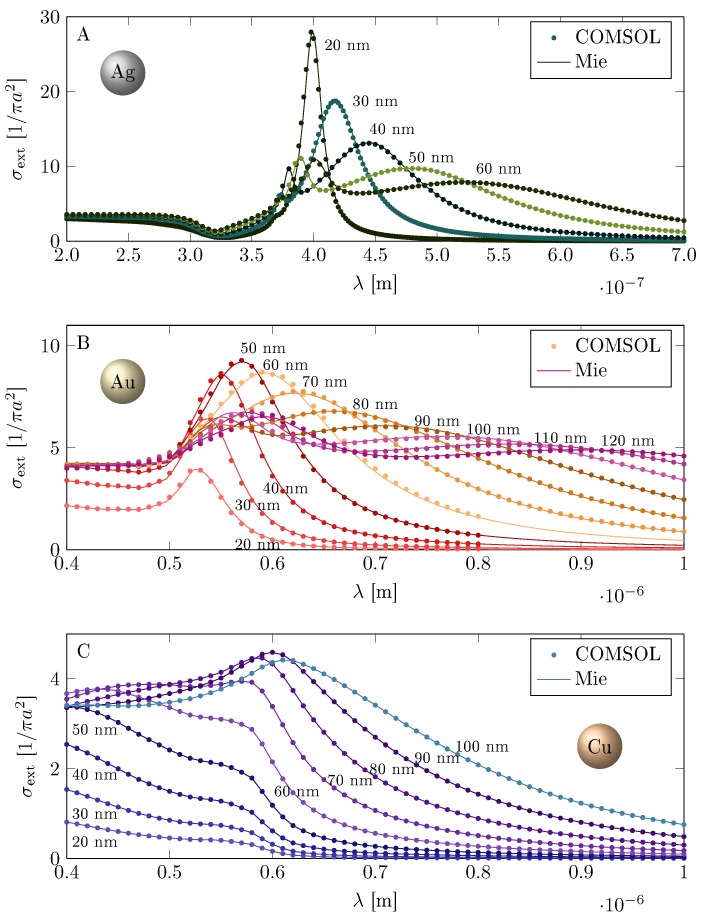
Surface plasmon resonances (extinction cross-section $\sigma_{ext}\left(\lambda \right)$ spectral dependence) for Au (**A**), Ag (**B**) and Cu (**C**) spherical nanoparticles with various radii. The dominating peak corresponding to the dipole plasmon mode at the radii region a∈ (10–100) nm overlaps very well with the visible part of the sun-light spectrum, which is convenient for plasmon-photovoltaic applications.

**Figure 3 materials-12-03192-f003:**
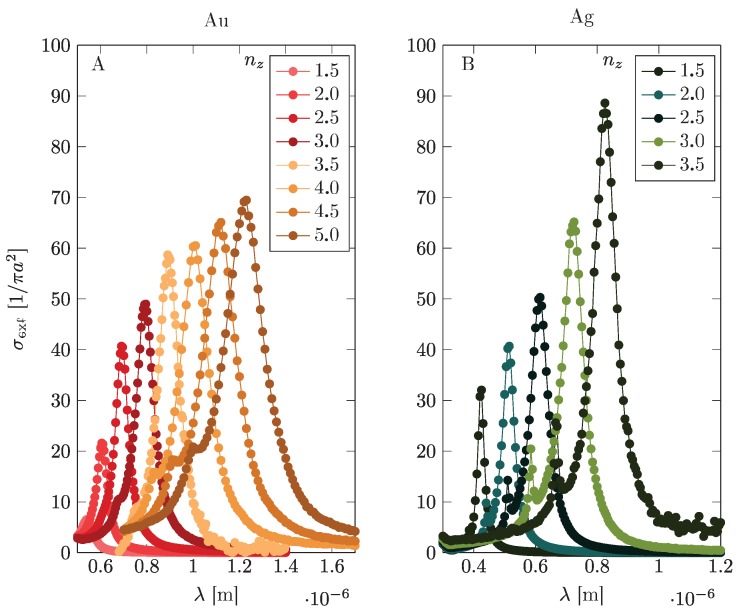
The dependence of the surface plasmon resonance (extinction cross-section spectral dependence) with respect to the permittivity $\varepsilon =n_{z}^{2}$ of the dielectric surroundings ($n_{z}$ is the refraction index of the dielectric medium, as marked in the figure). The dependence is relatively strong and in the figure shown for Au and Ag nanoparticles with radius a=20 nm.

**Figure 4 materials-12-03192-f004:**
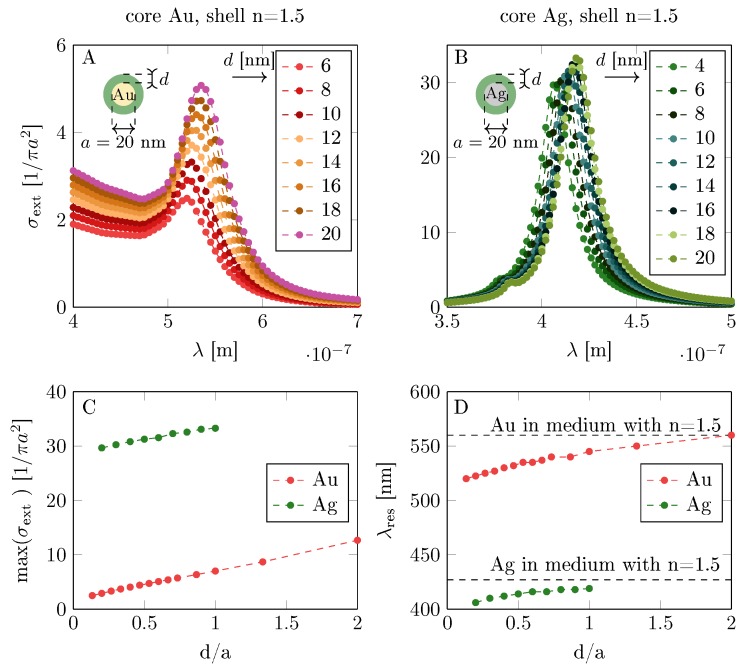
Red shift of the surface plasmon resonance (extinction cross-section spectral dependence) in metallic spherical core with respect to the thickness of the dielectric coating layer, *d* (for core Au (**A**), Ag (**B**)). In the figure, the size dependence is presented for Au and Ag spherical cores with constant radii a=20 nm with respect to the coating thickness *d* with the dielectric of permittivity $\varepsilon =n^{2}=2.25$—maximal extinction cross-section (**C**) and central resonance wave length (**D**).

**Figure 5 materials-12-03192-f005:**
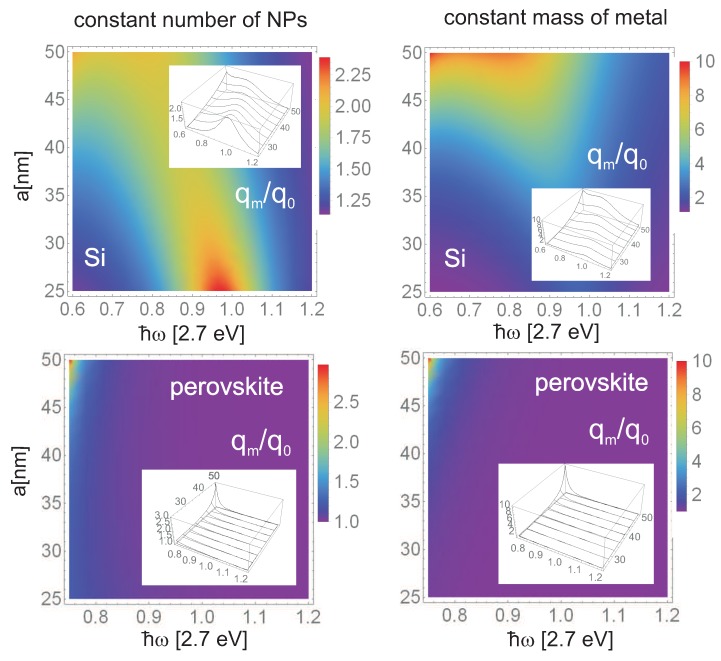
Spectral dependence of the efficiency increase $\frac{q_{m}}{q_{0}}$ (the ratio of probabilities of inter-band transitions per single photon with energy ℏω, with and without metallic nanoparticles (NPs)—according to Equation (Equation 17) and data listed in Table 1, Table 2 and Table 3) for Si cell (upper) covered with bare Au NPs with radii a∈ 25–50 nm. Similar spectral dependence of the efficiency increase $\frac{q_{m}}{q_{0}}$ for perovskite cell (lower) with core–shell NPs of the same size as for Si samples. Left panels correspond to a concentration of NPs kept constant when their size is varying, whereas the right panels correspond to the total mass of all metallic components kept constant. The significant difference is noticeable—for perovskite cells, the metallic NPs almost do not enhance the efficiency contrary to the Si cell, which is the result of inconveniently large effective masses of carriers in the perovskite and raises its forbidden gap in comparison with Si, entering the relation (Equation 17).

**Figure 6 materials-12-03192-f006:**
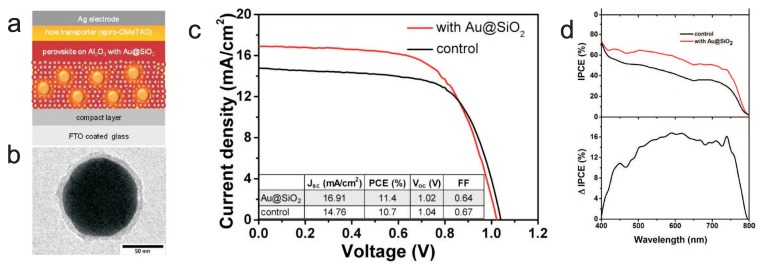
(**a**) illustration of device structure with the component-layers indicated (**b**) TEM image [13] of Au/SiO${}_{2}$ core–shell NP (b∼40 nm, a∼48 nm—cf. Figure 1); (**c**) I–V curve for perovskite cell with porous Al${}_{2}$O${}_{3}$ basis filled with Au/SiO${}_{2}$ core–shell nanoparticles with density ∼1 wt %; a strong increase of the photo-current is noticeable with simultaneous lowering of the voltage; (**d**) IPCE (photon-to-current conversion efficiency) spectra of control and with Au/SiO2 devices and its increase (ΔIPCE) due to the addition of Au/SiO${}_{2}$ NPs (devices measured under AM1.5 simulated sunlight, 100 mW/cm${}^{2}$ irradiance). After experiment [13].

**Figure 7 materials-12-03192-f007:**
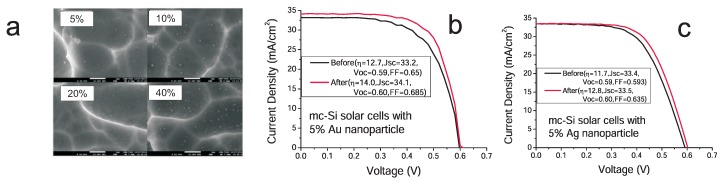
(**a**) EM photography [3] of distribution of metallic nanoparticles (Au) on the surface of the multi-crystalline Si solar cell—very low surface concentration of metallic components is visible (percentage refers to concentration of colloidal solution used for spin coating of a cell surface). (**b,c**) comparison of solar cell efficiency due to plasmon modification of the multi-crystal Si solar cell expressed by an increase of the area in the figure beneath the red I–V curve with respect to the area ranged by the black curve, (**b**) with Au nanoparticles (efficiency increase is ca. 6.5%), (**c**) by Ag nanoparticles (efficiency increase is ca. 2.5%), acc. to our study [3]. The significant difference in I–V characteristic for metallized Si and perovskite cells is noticeable—compare Figure 6c—which corresponds to different mechanism of plasmon induced efficiency increase, by strengthening of the photo-current due to easier dissociation of excitons in metallized perovskite cells.

**Table 1 materials-12-03192-t001:** Plasmon energies measured in metals.

Metal	Bulk pl [eV]	Surface pl in NPs [eV]
Li	6.6	3.4
Na	5.4	3.3
K	3.8	2.4
Mg	10.7	6.7
Al	15.1	8.8
Fe	10.3	5.0
Cu	6	3.5
Ag	3.8	3.5
Au	4.67	2.7

**Table 2 materials-12-03192-t002:** Mie frequency $\omega_{1}$ to Formula (Equation 17).

Metal	Au	Ag	Cu
Mie frequency	$4.11\times 10^{15}$ 1/s	$5.2\times 10^{15}$ 1/s	$5.7\times 10^{15}$ 1/s

**Table 3 materials-12-03192-t003:** Substrate material parameters to Formula (Equation 17) ($m=9.1\times 10^{-31}$ kg, the mass of bare electron; lh–light holes, hh–heavy holes, L–longitudinal, T–transverse).

Semiconductor	$m_{n}^{*}$	$m_{p}^{*}$	$E_{g}$
perovskite CH${}_{3}$NH${}_{3}$PbI${}_{3}$	1m	1m	1.6 eV
Si	0.9m L[101], 0.19m T[110]	0.16m lh, 0.49m hh	1.12 eV
GaAs	0.067m	0.08m lh, 0.45m hh	1.35 eV
CIGS	0.09–0.13m	0.72m	1–1.7 eV

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
