# Peer review of "Application of Core–Shell Metallic Nanoparticles in Hybridized Perovskite Solar Cell—Various Channels of Plasmon Photovoltaic Effect"

_materials, 2019, doi:10.3390/ma12193192_

Round 1
Reviewer 1 Report
Within the manuscript the authors reported a competently researched and reported work on semiconductor solar cells efficiency increase mediated by the surface plasmons in metallic nanoparticles.
The study focused on a new channel of plasmon photovoltaic effect related to the influence of plasmons onto the internal cell electricity and the simulation of plasmon increase of the efficiency of chemical perovskite has been compared with experimental observations.
The manuscript is clear, well arranged, the experimental and theoretical results are well presented. The research is scientifically sound, and the motivation is clear and the results are useful for deeper understanding of high efficient of semiconductor solar cells.
Author Response
Thank you very much for the positive opinion.
Reviewer 2 Report
Comments:
In this manuscript, Jacak et al. analyzed the microscopic mechanism of plasmon photovoltaic effect to improve the performance of perovskite solar cells by plasmons in metallic components. They found that the plasmon effect is a new way to increase the efficiency of perovskite solar cells by using metallic components which improve not only the absorption of photons but also the local electricity inside the device. The work has high novelty and impact, rigorous thinking and good writing. Thus, I can recommend the paper published in Materials in the present form.
Author Response

(The authors gave the same response as above.)

Reviewer 3 Report
Authors demonstrated a scheme for an energy transfer channel for surface plasmonic oscillations in metal nanoparticles with semiconductor absorption layer in perovskite solar cells. Here, authors analyzed the perovskite solar cells with CH3NH3PbI3−αClα active layer in which core shell nanoparticles Au-SiO2 embedded in the pores of ETL layer (i.e. TiO2 or Al2O3). Based on the application of Fermi golden rule, they found that the increment in Perovskite solar cell performance is not depends on absorption enhancement. Instead, the increased photo-current and 16% higher efficiency are attributed to the nanoparticle mediated reduced exciton binding energy of perovskite photo-excitons. The observed results were also compared with experimental results.
This works looks interesting and can be accepted after a minor revision based on following corrections.
The English and flow of the manuscript should be checked. For example, in conclusion line 244, perovskite mentioned as ‘parovskite’ and in line 247, Au/SiO2 mentioned as Au/SiO3. Check and correct in other places too. Core shell nanoparticles found to reduce the exciton binding energy of Perovskite, is the nanoparticle addition can alter the work function of ETL layer? In plasmonic enhancement, the distance between exction and nanoparticle surface is important to realize energy transfer, how we can account the distance between them in here? Can this scheme address the halide ion movement in perovskite? Which is a prime reason for instability? Could this scheme remain similar for planar perovskite solar cells? It should be mentioned or discussed in the manuscript.Author Response
Dear Referee,
Thank you very much for the opinion and recommendations. According to these recommendations,
The English and flow of the manuscript should be checked. For example, in conclusion line 244, perovskite mentioned as ‘parovskite’ and in line 247, Au/SiO2 mentioned as Au/SiO3. Check and correct in other places too. Core shell nanoparticles found to reduce the exciton binding energy of Perovskite, is the nanoparticle addition can alter the work function of ETL layer? In plasmonic enhancement, the distance between exction and nanoparticle surface is important to realize energy transfer, how we can account the distance between them in here? Can this scheme address the halide ion movement in perovskite? Which is a prime reason for instability? Could this scheme remain similar for planar perovskite solar cells? It should be mentioned or discussed in the manuscript.
we have made the following improvements in the submission,
The linguistic correction has been done and some improvement of English language has been performed. Many typos have been removed. Thank you very much for the identification of two, in line 244 (‘parovskite’ – it has been corrected to perovskite) and in line 247 (‘Au/SiO3’ – it has been corrected to Au/SiO2). Other typos (quite large number) has been also found and corrected. The question you have posed are very serious and extensive. As a matter of fact they touch large class of issues highly exceeding the goal of our submission. But all they are very important and we try to address to all of them in the discussion we have placed on the end of paragraph 4 (before Conclusions). The addition has the following form,“The hybrid perovskite cells differ considerably than conventional $p-n$ junction-type cells, like Si-based cells or other semiconductor cells (CIGS, GaAs and other). In latter cells photovoltaic excitons are instantly dissociated in the junction region, because of relatively high junction voltage (e.g., in Si of ca. 0.7 V) readily overcoming the binding energy of excitons. Separated there electrons and holes travel opposite directions to electrodes. Another situation is in exciton-types cells, chemical, plastic or perovskite ones, which operate without any $p-n$ junction and the dissociation of localized or itinerant excitons undergoes on the charge absorber surface (for electrons \cite{perow1} or holes \cite{perow2}) operating due to the difference of the bend edges at the interface between photo-active material and absorber. In chemical dye cells photo-active molecules are excited by incident photons and electrons outflow from these molecules to adjacent large gap semiconductor, TiO$_2$, due to the convenient slope of energy levels at the interface. Important is here the close contact of dye molecules with large effective surface of therefore porous TiO$_2$. The electrolyte restores next the initial state of dye by the re-dox cycle. In perovskite cells the photo-active material is the mid forbidden gap (ca. 1.6 eV) semiconductor CH$_3$NH$_3$PbI$_3$ (I can be substituted by Cl, which slightly changes elementary cell dimension and bend parameters as well as the chemistry of layer preparation) in which excitons are created by incident photons or by plasmons mediating sun-light absorption. These excitons dissociate at the interface with large gap semiconductor, TiO$_2$ or Al$_2$O$_3$ (being the electron absorbers due to the convenient slope of conduction band edges at the interface with the perovskite). Very important role plays the time of the diffusion of excitons to the active interface, because too long diffusion increases recombination of exciton reducing eventual photo-current. To struggle with such parasite effect the thin perovskite layers are applied (optimal are of ca. 290 nm of depth \cite{perow2}) and there are constructed porous bases for the perovskite layer increasing the surface and range of the interface. All this is the subject of optimization and trade-off of various competing factor, including also metallization by plasmon nanoparticles reducing exciton binding energy and, in this way, accelerating their dissociation and increasing eventual photo-current. To optimize plasmon influence on the dissociation process, the localization of metallic nanoparticles in a close vicinity of the interphase is convenient. However, the great obstacle on the way of practical utilization of perovskite cells, invincible yet, is their chemical instability and a short time of degradation if exposed to atmospheric conditions. A protection by glass or plastic layers complicates technology and is in practice insufficient to assure durability comparable to Si cells. Various methods are proposed to increase the durability of perovskite cells, e.g., fluorination by the plasma treatment \cite{perow6}. Perovskite cells exposed to fluorination process showed very good stability against water, light, and oxygen without any impact on the morphology and electrical property of the pristine cell. Such advances in preparation techniques show that perovskite cells might be a strong factor in the nearest future renewable green energy market.”The bibliography position [21] has been added,
Wu, C.; Wang, K.; Feng, X.; Jiang, Y.; Yang, D.; Hou, Y.; Yan, Y.; Sanghadasa, M.; Priya, S. Ultrahigh Durability Perovskite Solar Cells. Nano Lett. 2019, 19, 1251. doi: 10.1021/acs.nanolett.8b04778.
We once more express our gratitude for the comments and hope that we have corrected the submission in a satisfactory manner.
Sincerely yours,
Jacak (on behalf of authors)